# OpenReview forum: "Lexico: Extreme KV Cache Compression via Sparse Coding over Universal Dictionaries"
_ICML.cc/2025/Conference — ICML 2025 poster_

### Official Review · Reviewer_EhUD · 2025-03-12

**Overall Recommendation:** 3

**Summary:**

The paper introduces Lexico, a KV cache compression method using sparse coding over universal dictionaries. By leveraging Orthogonal Matching Pursuit for sparse approximation, Lexico provides flexible compression while maintaining high performance.

**Claims And Evidence:**

yes

**Essential References Not Discussed:**

nothing specific

**Experimental Designs Or Analyses:**

yes

**Methods And Evaluation Criteria:**

yes

**Other Comments Or Suggestions:**

nothing in particular

**Other Strengths And Weaknesses:**

Strengths:
1. The sparse dictionary learning for approximating KV cache compression appears new.
2. Extensive experiments are conducted. Lexico is tested on various LLMs, including Mistral-7B, Llama-3-8B, Llama-3.1-8B-Instruct, Llama-3.2-1B/3B-Instruct, and Qwen2.5-14B-Instruct. Lexico is benchmarked against prior KV cache compression techniques including Per-Token Q, ZipCache, PyramidKV, KIVI, and SnapKV and performs consistently better

Weakness:
1. The paper argues that key vectors lie in several low-rank subspaces, as demonstrated in Figure 2. However, it does not explicitly verify whether this property also holds for value vectors. It remains unclear whether the proposed dictionary-based sparse representation is equally effective for both keys and values.
2. A figure similar to Figure 2 for value vectors would strengthen the hypothesis by showing whether value vectors exhibit the same behavior.
The paper does not benchmark dictionary training time across different sparsity levels.

**Questions For Authors:**

Questions:
1. Would task-specific dictionaries improve performance? In Figure 6 for MMLU Pro Law, Lexico does not outperform competing methods. Is this due to the input distribution being significantly different from WikiText-103 where the dictionary was trained?
2. In Table 10, the computation only considers the key but does not include the value. Why is the computational cost for value vectors omitted in this analysis?
3. Please see Weakness.

**Relation To Broader Scientific Literature:**

yes

**Theoretical Claims:**

yes

---

> ### Author Rebuttal · Authors · 2025-04-01
>
> We thank you for recognizing the novelty of applying sparse dictionary learning for KV cache compression and our extensive experimental evaluation—a strength also highlighted by reviewers Vzmg and EtKf. We address each concern below.
>
> ***Q1: Task-specific dictionaries***
>
> Yes, using task-specific dictionaries may improve task accuracies, but would incur the additional training cost of custom fine-tuning. While we find this a promising direction to improve performance, our motivation to employ a universal dictionary was to allow it to be used off-the-shelf for various tasks without the need for additional training. That said, we can see that one may want to explore task-specific dictionaries to further enhance the performance of Lexico in specific domains.
>
> As MMLU-Pro Law does not have a training split, we tested task-specific dictionaries for GSM8K. Specifically, we present empirical results with the following dictionaries:
> 1. **Universal dictionary:** Dictionary trained on WikiText-103.
> 2. **Dictionary pre-trained on GSM8K:** Dictionary trained from scratch on the GSM8K training set.
> 3. **Universal dictionary fine-tuned on GSM8K:** Dictionary trained on WikiText-103 and then fine-tuned on GSM8K.
>
> | Sparsity | KV size | Universal dictionary | Dictionary pre-trained on GSM8K | Universal dictionary finetuned on GSM8K |
> | -------- | -------- | -------- | -- | -- |
> | $s = 24$     | 36.9%     | 76.88     | 77.63 | **79.68** |
> | $s = 14$     | 26.1%     | 75.06     | 70.51 | **75.89** |
> | $s = 10$     | 22.7%     | **69.67**     | 59.59	| 67.25 |
>
> From the experimental results, we observe that training a dictionary from scratch solely on GSM8K significantly underperforms compared to the universal dictionary, especially at low sparsities. This indicates overfitting to the small GSM8K training set, failing to generalize even to its test set. In contrast, fine-tuning a universal dictionary on GSM8K sometimes provides marginal improvements but adds the additional overhead of task-specific fine-tuning. Therefore, we may expect a slight increase in Lexico's performance on MMLU-Pro given a dataset with similar text distributions.
>
>
> ***Q2: Table 10 computation of keys and values***
>
> Apologies for the confusion. To clarify, the reported latency for the Lexico forward pass in Table 10 represents the entire process of generating one token. The row stating "Lexico: forward pass using $q (K_\text{csr}D_k^\top ) ^\top$" should be corrected to "Lexico: forward pass using $q (K_\text{csr}D_k^\top ) ^\top$ and $V_\text{csr}D_v^\top$", as the overall latency measurement indeed includes the additional compute for value vector reconstruction. We will revise the manuscript to explicitly note that the Lexico forward pass timing encompasses both the key and value reconstruction computations.
>
> ***W1: Do value vectors also lie in low-rank subspaces?***
>
> Yes, our findings show that value vectors also lie in a low-rank subspace. We will include an analogous figure as Figure 2 for value vectors. Below we computed the average relative reconstruction errors over 1k tokens from the WikiText test split using a universal dictionary of size $N=4096$.
>
> | sparsity | $s=4$ | $s=8$ | $s=12$ | $s=16$ |
> | -------- | -------- | -------- | -------- | -------- |
> | Keys     | 0.158     | 0.091     |0.058     |0.038     |
> | Values   | 0.410     | 0.264     |0.171     |0.110     |
>
> Although the sparse approximation for values is less effective than for keys, we believe that our empirical results on real-world benchmarks (LongBench, GSM8K) clearly demonstrate the effectiveness of our dictionary-based sparse representation.
>
> ***W2: Dictionary training time across different sparsity levels***
>
> Training dictionaries is cheap because they only need to be trained once and can be provided by a third-party. As we report in L203, a sparsity of s=32 and a dictionary size of N=1024 requires only 2 hours on an A100 GPU. We provide the training runtime for different sparsities and dictionary sizes for Llama-3.1-8B-Instruct below and will incorporate these results in a revised edition.
>
>
> | - | $s=4$ | $s=8$ | $s=16$ | $s=32$ |
> | -------- | -------- | -------- | -------- | -------- |
> | $N=1024$     | 37m     | 52m     | 1h 10m | 1h 59m |
> | $N=4096$     | 1h 18m     | 1h 40m     | 2h 40m | 5h 22m |

---

### Official Review · Reviewer_fiw3 · 2025-03-14

**Overall Recommendation:** 4

**Summary:**

LLMs that process long contexts need to store token embeddings in a KV-cache: two matrixes K and V.
The KV cache can grow arbitrarily large, and thus should be compressed.
Lexico compresses the KV-cache using sparse coding where K is decomposed as D * S where D is a dictionary matrix of fixed size and S a sparse matrix (and similar for V)
Lexico uses a fixed dictionary D trained offline and offers a heuristic algorithm to estimate S (computing the optimal S is NP-hard).

## update after rebuttal

I read the other reviewer comments and the rebuttals. Although the authors did not do some requested experiments, I think the paper has enough merit to be published with the current experimental body.

**Claims And Evidence:**

see below

**Essential References Not Discussed:**

None

**Experimental Designs Or Analyses:**

Experiments are sufficient.
The latency should go into the main paper IMO.

**Methods And Evaluation Criteria:**

The evaluation seems fine.

**Other Comments Or Suggestions:**

There are some paragraphs that have been compressed to fit in 8 pages which looks ugly.

The results for different LLMs are largely redundant, so others could be moved to appendix

The OMP algorithm and the latency numbers *are* important so should be in the main text.

Detailed comments:

L149 what does overcomplete mean?

sec 2: There is one KV cache per head, while the exposure mentions one per layer all the time. It is unclear if the compression is performed for all heads of one layer (considering the keys and vals are of size d) or separately for each head (keys and vals of size m)

Fig 2: unclear what's on the x and y axis of these plots. IIUC 49 tokens from the same input sequence? Again is this for the whole layer (size d) or for one head of layer 10 (size m)

L198 the expectation maximization algorithm is described in the caption of fig 3 -- move to the main text

Tab 1: it is unclear what the error is measured on (and what the standard deviation refers to)

L216 right: the implementation uses a compressed domain multiplication, which is an interesting optimization that the authors could emphasize more

L230 right: AFAICS, n_a has not been introduced (it's described in the supp material)

**Other Strengths And Weaknesses:**

S1 the first application of sparse coding to KV-cache compression

S2 properly optimized: computation in the compressed domain and very cheap sparse encoder

S3 easy-to-read paper

W1 The paper is sometimes organized in a weird way -- see detailed comments below

W2 judging from algorithm 1 (in the appendix) the OMP sparse encoding is a greedy approximation of sparse coding. Since it is an approximation to a NP-hard problem, there should be a tradeoff between encoding time and accuracy. It would be interesting to see how the same algorithm would perform with eg. a more or less large beam size of encodings

**Questions For Authors:**

The dictionary training algorithm (caption of fig 3) is an EM that alternates between encoding the training set and estimating the dictionary. Estimating the dictionary can be done in closed form (linear least squares to minimize the reconstruction error). Why do the authors perform gradient steps instead ? Does it perform better?

**Relation To Broader Scientific Literature:**

The related work section is very short and not properly related to Lexico.
How does sparse coding compare to quantization methods like eg. Product Quantization?

**Theoretical Claims:**

there are none

---

> ### Author Rebuttal · Authors · 2025-04-01
>
> Thank you for your thoughtful review. We are glad that you appreciated the novelty of Lexico, our effort in optimizations, and the overall readability of the paper. We address the remaining weaknesses and questions below.
>
> ***W1: Paper organization***
>
> We greatly appreciate the reviewer’s feedback regarding the organization of our paper. In our revision, we plan to:
> - Reorganize the Layout: We will reposition diagrams and tables so that they enhance readability, moving tables to the top of pages rather than interspersed between paragraphs. This can be done without difficulty in the case of acceptance given the additional ninth page.
> - Move Mistral results for readability: We will move the Mistral 7B results to the supplementary material, addressing concerns of redundancy while still providing experiments sufficient for our message. The original intent was to demonstrate that Lexico is not tailored to any particular model.
> - OMP Algorithm and Latency Measurements: We agree that these are important and will adjust the layout to fit them in the main text.
>
> ***W2: Tradeoff between encoding time and accuracy***
>
> We compared OMP with alternative approaches—Iterative Hard Thresholding (IHT) and Lasso—on the WikiText test set using our universal dictionary (size N=4096). While IHT and Lasso are iterative methods that can achieve better accuracy with additional iterations (trading off computation time for improved performance), OMP’s greedy nature does not allow for such tuning. In our experiments, even after increasing the number of iterations (and thus compute time) for IHT and Lasso, they still produced higher reconstruction errors than OMP across various sparsity levels. Moreover, OMP was the fastest since its complexity scales with the sparsity s, which in our case is much smaller than N.
>
> |Sparsity|OMP|IHT|Lasso|
> |-|-|-|-|
> |s=4|0.28|0.81|0.57|
> |s=8|0.11|0.57|0.43|
> |s=16|0.05|0.35|0.28|
> |s=32|0.01|0.28|0.16|
>
> If the reviewer is aware of alternative approaches that may offer better tradeoffs, we would be eager to explore them.
>
> ***Q1: Estimating the dictionary via least squares***
>
> Indeed a closed-form solution via least squares exists in each update. However, this approach proved prohibitively slow for large training sets. The solution requires forming and inverting an $N \times N$ matrix, an operation that scales on the order of $\mathcal{O}\left ( N^3 \right )$. With $N = 4096$ in our experiments, such step quickly dominates overall runtime and overshadows any advantage of jumping straight to the global minimizer.
>
> ***Related work: Comparing sparse coding to product quantization***
>
> Thank you for the suggestion. We will add the reference and comparison to our Related Work. Product quantization and sparse coding both use a form of "codebook" for vector compression. Sparse coding represents each vector as a sparse linear combination of learned dictionary atoms, while product quantization partitions vectors (e.g., the first k coordinates) and clusters each sub-vector, concatenating the nearest centroids. PQCache (https://arxiv.org/abs/2407.12820) learns its centroids from input tokens; our method trains the dictionary offline. Despite these differences, both exploit the key vectors’ clustering structure for significant compression.
>
> ***L149: overcomplete definition***
>
> It means that the dictionary has more atoms (number of vectors) than the dimensionality of the vectors. This allows multiple ways to represent the same signal, making it easier to find a good sparse representation.
>
> ***Clarification on head-level compression***
>
> The compression is performed on key vectors of size $m$ (not $d$). The "one per layer" phrase in the paper refers to the dictionary rather than the KV cache. For each token, we have $L \times H$ key vectors of size $m$, while we have $L$ key dictionaries of shape $m \times N$. Thus, all $H$ heads within a layer share the same dictionary. An analogous setup applies to the value cache and dictionary.
>
> ***Figure 2 axes***
>
> In Figure 2, each axis represents 7 tokens across all 8 heads of layer 10, yielding 7×8=56 elements (with some elements cropped for clarity). The plotted values are from key vectors for each head (with dimensionality m). In the left panel, both axes correspond to the first 7 tokens from the same sentence, whereas in the right panel each axis represents the first 7 tokens from two different sentences. The elements are sorted by similarity to highlight clustering.
>
> ***Table 1 error***
>
> We measure the relative reconstruction error (RRE) for each key or value vector as $\text{RRE}=\frac{\|\mathbf{k}-\hat{\mathbf{k}}\|_2}{\|\mathbf{k}\|_2}$, where $\mathbf{k}$ is the original vector and $\hat{\mathbf{k}}$ its reconstruction. In Table 1, each reported value is the average RRE across 8k key/value vectors—corresponding to 1k tokens, each producing 8 head vectors—obtained from forward passes on the specified datasets. The standard deviation is computed over those 8k samples.

---

> > ### Comment · Reviewer_fiw3 · 2025-04-04
> >
> > W2: About the tradeoff experiment: in Algorithm 1, the encoding is greedy, ie. at each step i, the dictionary entry that maximizes the dot product with the current residual is selected.
> >
> > A straightforward extension is to keep not the single max but a set of top-B elements (the beam) and compute the encoding using these B elements. After s iterations, only the best out of B encodings is kept. This is guaranteed to give a better encoding since the top-1 is included in the top-B.

---

> > > ### Author Response · Authors · 2025-04-09
> > >
> > > Many thanks for clarifying your suggestion on extending OMP. We agree that selecting the top-B atoms at each iteration is guaranteed to provide a better (or at least no worse) sparse representation.
> > >
> > > Our primary motivation for using standard OMP in this work lies in its simplicity and efficiency for memory-constrained settings. While a beam search would likely improve reconstruction performance, it comes with additional computation and memory overhead proportional to the beam size. Moreover, in many cases, greedy OMP has well-established theoretical guarantees and performs near-optimally, especially when the dictionary is highly overcomplete, as in our case (Donoho et al., 2006, https://doi.org/10.1109/TIT.2005.860430; Elad, 2010, *Sparse and Redundant Representations: From Theory to Applications in Signal and Image Processing*, Theorem 4.3).
> > >
> > > Nonetheless, we appreciate your suggestion and are interested in exploring how beam search extensions might further enhance OMP’s reconstruction accuracy. We will include a discussion of this direction in the revised manuscript.

---

### Official Review · Reviewer_EtKf · 2025-03-15

**Overall Recommendation:** 3

**Summary:**

This paper proposes a novel KV cache compression method based on sparse coding over a learned “universal” dictionary. The key idea is to represent KV cache vectors with a small number of dictionary “atoms,” reducing memory costs while preserving model performance across different tasks and domains. The paper reports encouraging results on select language tasks (e.g., GSM8K, LongBench), demonstrating near-lossless performance at significant compression ratios.

## update after rebuttal

The authors have addressed most of my questions. I keep my opinion that this paper leans toward being accepted. The additional results and clarifications should be integrated into the manustript.

**Claims And Evidence:**

The author makes the following claims:
- Near-lossless performance: Supported by performance experiments (but only with the most safe settings)
- Compression rates beyond 2-bits: Partially supported by theoretical KV cache size analysis (no hardware peak memory report)
- Universality: Supported by experiment results across different datasets

**Essential References Not Discussed:**

N/A

**Experimental Designs Or Analyses:**

Experiment design is valid.

Need to report peak memory and throughput/latency to show efficiency.

**Methods And Evaluation Criteria:**

Overall, make sense.

Comments on evaluation datasets:
- Use Longbench V2
- Consider long output tasks such as reasoning

**Other Comments Or Suggestions:**

N/A

**Other Strengths And Weaknesses:**

I summarize all strengths and weaknesses here.

Strengths
- This method is intuitive and flexible, allowing fine-grained trade-offs by adjusting sparsity.
- The performance is good; this method often achieves near-lossless performance on standard benchmarks.
- The dictionary-based approach is an interesting alternative to quantization or token eviction, and the universality claim is a advantage.

Weaknesses
- The paper lacks real hardware measurements (latency, throughput) to show efficiency benefits.
- Each model needs its own dictionary, potentially adding overhead in multi-model or highly specialized scenarios.
- The authors do not benchmark on more advanced long-form tasks like MATH or AIME, where extended CoT generation can have more burden on memory.

**Questions For Authors:**

See weaknesses.

**Relation To Broader Scientific Literature:**

This work is about KV cache compression. Related to literatures like quantization, token eviction, etc.

**Theoretical Claims:**

N/A

---

> ### Author Rebuttal · Authors · 2025-04-01
>
> We thank the reviewer for the thoughtful and constructive feedback. We are pleased that you appreciated Lexico’s flexible design, near-lossless performance on benchmarks, and the advantage of using a dictionary for KV cache compression. It is also encouraging that reviewers Vzmg and EhUD similarly highlighted our method's strong performance, particularly regarding its improved memory vs. performance trade-offs compared to other compression methods. We address the weaknesses and questions below.
>
> ***Hardware measurements***
>
> Section E of the paper already presents a latency analysis and, hence, claiming we "lack real hardware measurements" is misleading. In light of the reviewer's emphasis on this point, we will move that section into the main body in the revised manuscript. We have also extended the same setup described in Section E to include varying sparsity levels for a more fine-grained analysis; these results will be incorporated into the revised edition.
>
> |||N=1024||||N=4096||||
> |-|-|-|-|-|-|-|-|-|-|
> ||Full KV|s=4|s=8|s=16|s=24|s=4|s=8|s=16|s=24|
> |**Forward pass**|48.39 ms|53.90 ms|54.67 ms|55.46 ms|55.56 ms|57.24 ms|56.64 ms|57.13 ms|56.35 ms|
> |**Sparse approximation via OMP**|-|6.03 ms|10.16 ms|18.31 ms|26.57 ms|9.37 ms|15.55 ms|28.37 ms|40.58 ms|
>
> Although Lexico may incur higher latency at larger sparsities, it is designed to address memory-constrained scenarios where a modest latency trade-off is acceptable, and hardware or library optimizations (e.g., support for FP8 multiplications) can further narrow this gap while preserving substantial memory savings.
>
> ***Lexico for CoT/reasoning tasks***
>
> We are glad the reviewer noted the potential for Lexico in long-form reasoning tasks. To investigate this, we tested a reasoning model, DeepSeek-R1-Distill-Llama-8B, on the AIME 2024 exam. We evaluated the performance trade-off when using Lexico to compress the KV cache memory of long reasoning traces:
>
> |Compression method|KV Size|AIME 2024 (pass@1)|
> |-|-|-|
> |Full KV Cache|100%|30.0|
> |KIVI (4 bits)|31.8%|30.0|
> |Lexico ($s=24$)|28.6%|30.0|
> |KIVI (2 bits)|16.3%|23.3|
> |Lexico ($s=14$)|17.0%|26.7|
>
> At a moderate compression level ($s=24$), Lexico retains full-precision performance (30%). When the compression becomes more aggressive, as with Lexico ($s=14$) and KIVI (2 bits), the model's score declines, though Lexico still scores higher than KIVI at similar compression levels.
>
> One approach to improve performance is to train task-specific or reasoning-specific dictionaries. Recent reasoning models generate traces that extend well beyond the prompt, so much of the KV cache is used for these model-generated traces. Because these traces may inhabit a different subspace than the prompt KV cache, training the Lexico dictionary specifically on output generations could be a promising direction for improving the performance-memory trade-off in long-form reasoning tasks. We demonstrate this potential for task-specific dictionaries and relevant ablations in the "Dictionary sources and adaptability across domains" section in our response to reviewer Vzmg.

---

> > ### Comment · Reviewer_EtKf · 2025-04-04
> >
> > Thanks for sharing the additional results. Could you provide a throughput comparison under different batch sizes and sequence length settings? This information would be particularly useful because KV cache compression is especially beneficial when serving large batch sizes and longer sequences.

---

> > > ### Author Response · Authors · 2025-04-09
> > >
> > > We appreciate the reviewer's suggestion to include a throughput comparison. We would like to emphasize that our primary focus is on addressing extremely memory-constrained scenarios, where even a small batch size may exceed device limits. Such scenarios are increasingly relevant in real-world applications, especially with the emergence of reasoning models with lengthy CoT generation and LLM agents operating with very large contexts.
> > >
> > > Nevertheless, we acknowledge that throughput is an important consideration for many use cases. We will include throughput experiments in the revised manuscript to provide a more comprehensive evaluation of our approach. Thank you for highlighting this.

---

### Official Review · Reviewer_Vzmg · 2025-03-25

**Overall Recommendation:** 2

**Summary:**

Lexico compresses the KV cache by finding a sparse representation of each key and value vector using a pre-trained dictionary. Instead of storing the full high-dimensional vectors, Lexico approximates them as a sparse linear combination of a small number of “atoms” (basis vectors) from this dictionary. It uses a technique called Orthogonal Matching Pursuit to efficiently select only a few atoms that best reconstruct the original vector, minimizing the reconstruction error. These sparse representations are much smaller in size and can be stored efficiently using formats like CSR, reducing memory usage.

**Claims And Evidence:**

The submission's claims are generally well-supported by clear experimental results across multiple benchmarks and models. The evidence convincingly demonstrates Lexico’s compression effectiveness and performance retention. No major claims appear problematic.

**Essential References Not Discussed:**

The paper overlooks an essential reference: **QJL: 1-Bit Quantized JL Transform for KV Cache Quantization with Zero Overhead by Zandieh et al**. This work proposes a highly efficient 1-bit quantization method for KV cache compression, offering strong compression with minimal overhead. Given that Lexico aims to outperform quantization-based baselines, omitting QJL misses an important point of comparison—especially in the context of extreme low-memory regimes, where QJL sets a relevant benchmark. Including this citation would strengthen the discussion of related methods.

**Experimental Designs Or Analyses:**

Yes, the experimental design appears sound. The authors compare Lexico against strong baselines across diverse models and tasks, using consistent settings and fair memory budgets. The analyses of compression vs. performance trade-offs are thorough, and no major issues were found.

**Methods And Evaluation Criteria:**

Yes, the methods and evaluation criteria are appropriate for the problem. The use of sparse coding with universal dictionaries is well-justified, and the evaluation on standard benchmarks like LongBench and GSM8K effectively demonstrates performance and memory trade-offs.

**Other Comments Or Suggestions:**

-

**Other Strengths And Weaknesses:**

Weaknesses include a limited evaluation on LongBench only a subset of tasks is tested, which may not fully capture long-context challenges. The paper also lacks a "needle-in-a-haystack" style experiment, which is important for stress-testing memory compression in precision-critical scenarios. Additionally, while Lexico is positioned as efficient, more detailed analysis of training runtime, dictionary update cost, and latency during decoding (especially under different dictionary sizes and sparsity levels) would strengthen the claims. The absence of ablations on different dictionary sources or adaptability across domains is another gap worth addressing.

**Questions For Authors:**

-

**Relation To Broader Scientific Literature:**

The paper builds on prior work in KV cache compression, sparse coding, and dictionary learning. It extends ideas from compressed sensing and applies them in a novel way to LLM inference. Compared to quantization and eviction-based methods, Lexico offers finer-grained control and better performance in low-memory regimes, addressing limitations in previous approaches.

**Theoretical Claims:**

The paper does not present formal theoretical proofs, but relies on established techniques like Orthogonal Matching Pursuit and sparse coding. The theoretical framing is sound, and no correctness issues were identified in the described methodology.

---

> ### Author Rebuttal · Authors · 2025-04-01
>
> We are pleased that you appreciated the clarity of our experiments, Lexico's effectiveness, and our use of sparse coding. EtKf and EhUD similarly recognized the soundness of our experimental design. We address your concerns below, and would be grateful if you reevaluate our work in light of them.
>
> ***LongBench***
>
> We do not agree that our evaluation is limited. We have covered a comprehensive set of task categories such as document QA, summarization, few-shot tasks, and code completion; this aligns with prior work (e.g., KIVI) that evaluated on the same subset. For completeness, we have Lexico on all LongBench tasks with Llama-3.1-8B-Instruct below.
>
> ||Avg.|NarrativeQA|Qasper|MultiFieldQA|HotpotQA|2WikiMultihopQA|MuSiQue|GovReport|QMSum|MultiNews|TREC|TriviaQA|SAMSum|PassageCount|PassageRetrieval|LCC|RepoBench-P|
> |-|-|-|-|-|-|-|-|-|-|-|-|-|-|-|-|-|-|
> |Full KV|39.02|23.43|22.54|28.42|16.63|15.3|9.05|35.04|24.57|27.44|72.5|91.65|43.47|1.01|93.33|63.15|56.76|
> |s=16|38.33|23.02|15.45|28.62|16.39|13.8|10.56|31.36|23.13|25.78|72.5|92.25|42.02|1.47|98.33|63.01|55.58|
> |s=8|34.23|14.29|11.66|22.04|13.89|12.45|9.74|23.41|21.04|22.35|60.0|91.01|40.30|0.89|93.62|59.60|51.46|
>
> The newly added tasks demonstrate a similar trend as the other tasks. The performance remains nearly lossless when $s=16$ (21% of the original KV cache size). Our results remain consistent with our subset of tasks in Table 2. Even at $s=8$ (12% of the original), our approach maintains a strong performance-memory tradeoff; note that 12% is a regime 2-bit quantization cannot achieve. These results show Lexico's strength in handling long contexts.
>
> ***Needle In A Haystack (NIAH)***
>
> We show that Lexico is resilient to such stress-testing. Using Llama-3.1-8B-Instruct with an 8k context size, we followed the settings in PyramidKV. We compared against token eviction methods—which are well-suited for tasks that require only a few tokens from the context—and report ROUGE-1 multiplied by 10. Note that the columns represent the needle depth ratio.
>
> We find that Lexico outperforms competitors on a 20% KV cache budget, while it delivers similar performance on a 10% budget. Note that the 10% budget is not feasible for 2-bit quantization methods. While we believe that NIAH is a valuable synthetic benchmark, performance on the task does not strongly reflect that of practical performance, as exemplified in the underperformance of eviction-based methods on GSM8K (Figure 4).
>
> |Method|KV Cache|0.0|0.2|0.4|0.6|0.8|1.0|
> |-|-|-|-|-|-|-|-|
> |Full KV|100%|7.06|7.06|9.30|7.06|7.06|7.06|
> |Lexico$_{s=16}$|20%|**7.06**|**7.06**|**9.09**|**7.06**|**7.06**|7.06|
> |PyramidKV|20%|6.06|4.12|4.24|4.78|4.18|**7.41**|
> |SnapKV|20%|6.06|4.12|4.24|5.15|3.94|**7.41**|
> |Lexico$_{s=8}$|10%|3.87|3.61|**5.88**|4.48|**4.26**|4.75|
> |PyramidKV|10%|**6.06**|**4.78**|4.18|4.12|4.18|**7.55**|
> |SnapKV|10%|**6.06**|4.18|4.85|**4.62**|3.94|7.41|
>
> ***Dictionary training cost***
>
> Training dictionaries is cheap because they only need to be trained once and can be provided by a third-party. As we report in L203, sparsity of s=32 and a dictionary size of N=1024 require only 2 hours on an A100 GPU. We provide the training runtime for different sparsities and sizes for Llama-3.1-8B-Instruct.
>
> | |s=4|s=8|s=16|s=32|
> |-|-|-|-|-|
> |N=1024|37m|52m|70m|119m|
> |N=4096|78m|100m|160m|322m|
>
> ***Latency***
>
> We kindly direct the reviewer to the *Hardware measurements* section in our response to reviewer EtKf (R2).
>
> ***Dictionary sources and adaptability across domains***
>
> Per your suggestion, we investigated whether training a dictionary on a specific domain enhances the performance. In our ablation below, we trained a dictionary on different sources:
> 1) Dictionary trained on WikiText-103.
> 2) Dictionary trained on GSM8K training set.
> 3) Dictionary trained on WikiText-103 then finetuned on GSM8K.
>
> |Sparsity|KV size|Trained on WikiText|Trained on GSM8K|Finetuned on GSM8K|
> |-|-|-|-|-|
> |s=24|36.9%|76.88|77.63|**79.68**|
> |s=14|26.1%|75.06|70.51|**75.89**|
> |s=10|22.7%|**69.67**|59.59|67.25|
>
> We observe that training a dictionary from scratch on GSM8K significantly underperforms, especially at low sparsities. Overfitting to GSM8K fails to generalize even to its test set. In contrast, finetuning the universal dictionary on GSM8K provides marginal improvement but adds the additional overhead of custom fine-tuning.
>
> ***Missing reference***
>
> We appreciate the reviewer’s suggestion and will incorporate QJL into our related work. While QJL introduces a 1-bit quantization approach for KV cache compression, its experiments are confined to 3- and 5-bit schemes. In contrast, our work benchmarks Lexico against 2- and 4-bit state-of-the-art methods, and further pushes the efficiency by exploring memory regimes lower than 2 bits. Whereas QJL applies its transformation solely to keys (with per-token quantization for values), our approach compresses both keys and values, offering a more comprehensive and efficient solution.

---

### Decision · Program_Chairs · 2025-05-01

**Decision:**

Accept (poster)

**Comment:**

The paper proposes Lexico, a compression scheme of the KV cache by finding a sparse representation of each key and value vector using a pre-trained dictionary. Most of the reviewers were quite positive about the paper, given the method's significant compression rate while mostly maintaining the accuracy performance. The authors also provided several additional experimental results that corroborates the contribution of the paper. One reviewer (Vzmg) was somewhat critical about a missing reference, but given the overall contribution, the paper seems worth publishing at ICML, as also has been acknowledged by other reviewer (fiw3). So, the recommendation is Accept.